# *Plasmodium falciparum* 7G8 challenge provides conservative prediction of efficacy of PfNF54-based PfSPZ Vaccine in Africa

Joana C. Silva [1,2], Ankit Dwivedi[1], Kara A. Moser[1], Mahamadou S. Sissoko[3], Judith E. Epstein[4], Sara A. Healy [5], Kirsten E. Lyke[6], Benjamin Mordmüller [7,8,9], Peter G. Kremsner [7,8], Patrick E. Duffy [5], Tooba Murshedkar[10], B. Kim Lee Sim[10], Thomas L. Richie [10] & Stephen L. Hoffman [10 ✉]

Controlled human malaria infection (CHMI) has supported *Plasmodium falciparum* (Pf) malaria vaccine development by providing preliminary estimates of vaccine efficacy (VE). Because CHMIs generally use Pf strains similar to vaccine strains, VE against antigenically heterogeneous Pf in the field has been required to establish VE. We increased the stringency of CHMI by selecting a Brazilian isolate, Pf7G8, which is genetically distant from the West African parasite (PfNF54) in our PfSPZ vaccines. Using two regimens to identically immunize US and Malian adults, VE over 24 weeks in the field was as good as or better than VE against CHMI at 24 weeks in the US. To explain this finding, here we quantify differences in the genome, proteome, and predicted CD8 T cell epitopes of PfNF54 relative to 704 Pf isolates from Africa and Pf7G8. We show that Pf7G8 is more distant from PfNF54 than any African isolates tested. We propose VE against Pf7G8 CHMI for providing pivotal data for malaria vaccine licensure for travelers to Africa, and potentially for endemic populations, because the genetic distance of Pf7G8 from the Pf vaccine strain makes it a stringent surrogate for Pf parasites in Africa.

[1] Institute for Genomic Sciences, University of Maryland School of Medicine, Baltimore, MD, USA. [2] Department of Microbiology and Immunology, University of Maryland School of Medicine, Baltimore, MD, USA. [3] Malaria Research and Training Center, Mali National Institute of Allergy and Infectious Diseases International Centers for Excellence in Research, University of Science, Techniques and Technologies of Bamako, Bamako, Mali. [4] Malaria Department, Naval Medical Research Center, Silver Spring, MD, USA. [5] Laboratory of Malaria Immunology and Vaccinology, NIAID, NIH, Bethesda, MD, USA. [6] Center for Vaccine Development and Global Health, University of Maryland School of Medicine, Baltimore, MD, USA. [7] Institute of Tropical Medicine, University of Tübingen and German Center for Infection Research, Tübingen, Germany. [8] Centre de Recherches Médicales de Lambaréné, Lambaréné, Gabon. [9] Department of Medical Microbiology, Radboud University Medical Center, Nijmegen, The Netherlands. [10] Sanaria Inc., Rockville, MD, USA. ✉email: slhoffman@sanaria.com

According to the World Health Organization's annual World Malaria Reports[1], in 2020, there were an estimated 241 million cases and 627,000 deaths caused by malaria—little change compared to 2015[1]. More than 98% of all deaths from malaria are caused by *Plasmodium falciparum* (Pf) and >90% of cases and deaths from Pf occur in sub-Saharan Africa[1]. The actual number of deaths may be at least 50% higher[2,3]. To put the mortality data in context, more people die every 11 days from malaria than the 11,325 who died during the 2014–16 Ebola epidemic[4]. An effective malaria vaccine would be of enormous benefit to those living in malaria-endemic areas, especially in Africa, and especially against Pf [5].

Malaria is also a serious concern for travelers. Between 2000 and 2014 there were an average of 1469 hospitalizations and 11 deaths per year in the United States (US) from malaria[6]. In 2019, there were 1719 cases of imported malaria and 15 deaths reported in the United Kingdom (UK)[7]; in 2017 and 2018 there were 957 and 896 cases, respectively, in Germany (Robert Koch-Institut: SurvStat@RKI 2.0, h.s.r.d., inquiry date: 18JUL2019), and in 2016, 2017, and 2018, there were 8427, 8347, and 8349 cases, respectively, in the European Union (EU)[8]. The large majority of these cases were acquired in Africa. In 2014–2016, among all malaria cases diagnosed in the US, 82–86% were acquired in Africa; among Pf cases imported into the US, 95–98% originated in Africa[9–11]. In 2017, 93% of the cases of malaria in Germany with a known place of origin came from Africa[12]. An effective malaria vaccine would thus be of enormous benefit to travelers as well, especially a vaccine to prevent Pf infections in travelers to Africa[5].

We are developing malaria vaccines composed of aseptic, purified, cryopreserved Pf sporozoites (SPZ) from a West African parasite strain, PfNF54, called PfSPZ vaccines[13]. The vaccines are initially intended for use by individuals living in Africa, and for travelers to Africa, and eventually for use in mass vaccination programs (MVPs) to eliminate malaria from defined geographical regions in Africa. We have been using small trials, including trials with controlled human malaria infection (CHMI), to assess vaccine efficacy (VE) of PfSPZ vaccines to optimize dosage regimens before conducting larger, more time-consuming, and expensive field trials[14–25]. CHMI is a safe, widely used procedure in which individuals are inoculated with infectious parasites and followed for the development of parasitemia, with treatment instituted as soon as parasitemia is detected. Small trials have used both homologous CHMIs, in which the vaccine and CHMI strains are the same (both PfNF54 or a clone derived from it called Pf3D7), and heterologous CHMIs, in which the CHMI strain differs genomically and therefore antigenically from the vaccine strain[14–19,22,25–28].

Heterologous CHMI with Pf parasites from Brazil (Pf7G8)[29,30] has been more stringent in individuals immunized with PfSPZ vaccines than homologous CHMI, with lower VE, after immunization with identical vaccine regimens[17]. The differences in VE after homologous *vs.* heterologous CHMI are almost certainly due to differences in antigenic targets in heterologous *vs.* homologous Pf strains. Comparative genome analysis has shown that Pf parasites, defined by genome sequence, cluster by geographical regions, with isolates from Africa, South America, Asia, and Oceania each more similar to others from the same continent than to those isolated from the other regions[31]. Nonetheless, there are significant differences among parasites populations from different parts of the same continent, for example among those from East, Central, and West Africa[28]. Studies by other groups have underscored the effect of genetic divergence on VE against CHMI[32,33].

The relevance of CHMI stringency on the ability to use CHMI as a surrogate for field efficacy has never been clear, despite the tremendous benefit that would accrue through savings in time, expense, and numbers of human research subjects if the relationship could be defined. Being able to predict field efficacy through small CHMI trials could help with optimization of vaccine design, choosing the best route of administration and the optimal regimen, and down-selecting the best candidates without resorting to expensive field trials. Here we demonstrate, using our clinical data, that assessment of VE of our most advanced product, PfSPZ Vaccine (radiation-attenuated PfSPZ), in adults in the US against heterologous CHMI with a South American parasite (Pf7G8) is at least as, and perhaps more, stringent than an assessment of VE against field exposure to Pf in Malian adults immunized with the identical regimen[17,28,34,35].

To better understand this finding, we undertook a comprehensive analysis of genomic data from ~700 African isolates, the vaccine strain (PfNF54), and the challenge strain (Pf7G8), examining the degree of genetic, proteomic, and predicted immunomic (CD8 T cell epitopes) divergence. We found that Pf7G8 is more divergent from PfNF54 than any of 704 African isolates originating from all over the continent. We conclude that heterologous CHMI trials with this genomically and antigenically divergent, non-African strain should provide a reliable and conservative estimate of the protection expected in malaria-inexperienced travelers exposed to Pf in Africa. In fact, since we cannot test our vaccines against all the isolates from Africa in the field, it may be that assessing VE against CHMI with Pf7G8, which is more distant from the vaccine strain than all of the 704 Pf strains from Africa that we assessed, may be a much sought after surrogate for conducting field trials in Africa. By extending our argument, a similar rationale can be defined for developing effective PfSPZ vaccines for use in geographic areas in other regions of the world.

## Results

**Clinical data from two sets of PfSPZ Vaccine Immunization Regimens in Malaria-Unexposed and Malaria-Exposed Adults.** The first immunization regimen (immunization regimen #1) consisted of $2.7 \times 10^5$ PfSPZ of PfSPZ Vaccine administered at 1, 5, 9, 13, and 21 weeks (0, +4, +8, +12, and +20 weeks). In the Joint Warfighter Medical Research Program 1 (JWMRP1) clinical trial, in the US, in previously malaria-unexposed adults (malaria-naïves), VE was 80% (4/5) against heterologous CHMI with Pf7G8 at 3 weeks after the last dose of vaccine, and 10% (1/10) at 24 weeks[17].

In the Mali 1 clinical trial, in Mali, in previously malaria-exposed adults (semi-immunes), VE against first Pf infection was 52% by 1-hazard ratio and 29% by 1-risk ratio analysis against intense transmission of heterogeneous Pf during 24 weeks after the last dose of vaccine[34]. In this trial, the incidence rate of new Pf infections in the controls was 93% during 24 weeks after the last dose of normal saline placebo. Although the VE by 1-risk ratio analysis in Mali (29%) was higher than in the US, the 95% confidence intervals overlapped and thus, the differences were not statistically significant[17,34]. This was in part due to the small sample size in the CHMI study.

Antibody responses to PfSPZ Vaccine were much lower in subjects in Mali than in subjects in the US. Antibody levels (the serum dilution at which the optical density was 1.0) to Pf circumsporozoite protein (CSP), the major protein on the surface of PfSPZ, 2 weeks after the last dose of vaccine were 28 times higher in vaccinees in the USA (median OD 1.0 = 11,179) than in vaccinees in Mali (OD 1.0 = 393)[17,34].

In summary of data from these two clinical trials with immunization regimen #1, we observed that the VE against first Pf malaria (parasitemia) in Mali during 24 weeks was higher (52% by 1-hazard ratio and 29% by 1-risk ratio) than the VE against

heterologous CHMI with Pf7G8 at 24 weeks in the US (10% by 1-risk ratio), despite the fact that the antibody immunogenicity of PfSPZ Vaccine was 28 times lower in the Malians.

The second immunization regimen (immunization regimen #2) consisted of $1.8 \times 10^6$ PfSPZ of PfSPZ Vaccine administered at 1, 9, and 17 weeks (0, +8, and +16 weeks). In the JWMRP2 clinical trial, in the US, also in malaria naïve adults, VE was 23% (3/13) against heterologous CHMI with Pf7G8 at 24 weeks after the last dose of vaccine[28].

In the Mali 2 clinical trial, in Mali, again in semi-immune adults, VE against first Pf infection was 51% by 1-hazard ratio and 24% by 1-risk ratio analysis against intense transmission of heterogeneous Pf during 24 weeks after last dose of vaccine[35]. In this trial, the incidence rate of new Pf infections in the controls was 77.8% during 24 weeks after the last dose of normal saline placebo.

As in the Mali 1 trial, antibody responses to PfSPZ Vaccine were much lower in subjects in Mali than in subjects in the US. Antibody levels to PfCSP 2 weeks after the last dose of vaccine were 9 times higher in vaccinees in the US (OD $1.0 = 16{,}795$) than in vaccinees in Mali (OD $1.0 = 1811$)[28,35].

In summary of data from these two clinical trials with immunization regimen #2, we observed that the VE against first Pf malaria (parasitemia) in Mali during 24 weeks was higher or at least as good (51% by 1-hazard ratio and 24% by 1-risk ratio) as the VE against heterologous CHMI with Pf7G8 at 24 weeks in the US (23% by 1-risk ratio), despite the fact that the antibody immunogenicity of the vaccine was 9 times lower in the Malians.

**Interpretation of why CHMI with Pf7G8 was at least as rigorous a test of VE as intense field exposure, in light of genomics and proteomics data.** We hypothesized that the primary reason VE against CHMI with PfSPZ Challenge (Pf7G8) in malaria-naïve subjects has been as, or more, stringent than has VE against natural exposure in semi-immune subjects in Africa, is that the genome, and associated proteome and immunome, of Pf7G8 are genetically more distant from PfNF54, the strain used in PfSPZ Vaccine, than are Pf isolates from across Africa, as we detail below.

Our previous analyses of whole genome assemblies show that PfNF54 is nearly identical to its clone 3D7, the source of the Pf reference genome[31]. The two differ by 1383 single nucleotide polymorphisms (SNPs) (60 SNPs per million base pairs), of which 80 alter predicted amino acid residues, and only 17 change predicted amino acid residues in non-pseudogene, single-copy, protein-coding genes. In addition, the two genomes also differ by ~8400 insertion and/or deletion mutations (indels), of which <3% fall in protein-coding genes. The latter mostly results in expansion/contraction of microsatellite repeats. In contrast, the genetic distance between 3D7 (and, by proxy, to PfNF54) and Pf7G8, is substantial, with 43,859 SNPs, of which 15,490 result in amino acid changes, and >47,000 individual indels.

Here, we calculated pairwise genetic differences between PfNF54, the strain in PfSPZ Vaccine, and each of several hundred Pf clinical isolates from Africa, as well as between PfNF54 and Pf7G8. Genetic distance, $p$, between PfNF54 and each isolate was calculated as the proportion of all high-quality variable genomic sites in the dataset that differ between each strain and PfNF54, with all comparisons limited to positions genotyped in Pf7G8. The total number of high quality variable sites in the dataset was determined as follows: of the 23 Mb-long Pf reference genome, 1,005,408 bp sites passed stringent quality filters (see Methods). Elimination of sites difficult to genotype or containing possible artifacts (those with missing data in >10% of all samples, $n = 99{,}037$; rare alleles, present in at most two of the

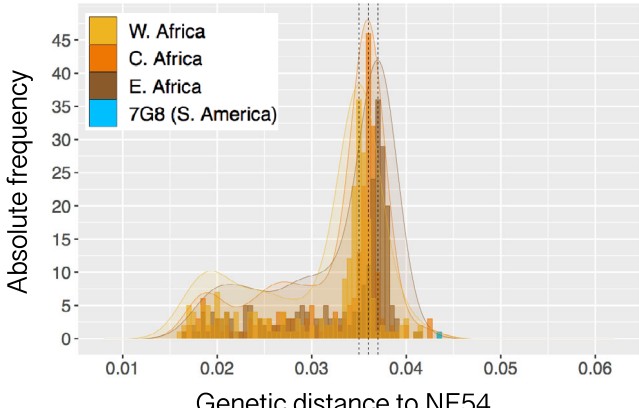

**Fig. 1 Pairwise genetic distance of 704 Pf African isolates and of Pf7G8 to NF54, the Pf strain in PfSPZ Vaccine.** Histogram shows the proportion of 137,585 variable sites that differ between Pf isolates from West (yellow), Central (orange), and East (brown) Africa and NF54 (X-axis). The Y-axis represents the count of isolates for each genetic distance. For improved visualization, a Gaussian kernel density plot is overlaid on the histograms (shaded). The increasing genetic distance, from West to East, of African isolates to NF54 is expected, since NF54 is thought to originate from West Africa[31]. The pairwise genetic distance between NF54 and the Brazilian clone 7G8 (blue) is larger than the distance of any African isolate and NF54.

>1000 global Pf isolates used in SNP calling, $n = 768{,}538$) reduced the number of variable sites analyzed to 137,833. Of these, 248 could not be called in Pf7G8. Therefore, the total number of variable sites analyzed was 137,585.

The distribution of $p$ distances between PfNF54 and Pf isolates from West, Central, and East Africa, plotted as a percentage of all variable sites that differ in each pairwise comparison, had modal values of 3.50, 3.57, and 3.70%, respectively. In each distribution, peaks with lower frequency than the mode were an artifact resulting from the algorithm of SNP calling in polyclonal samples (Supplemental information). Pf7G8 differed from PfNF54 more than any African isolate, with SNPs at 4.35% of all variable nucleotide sites (Fig. 1). Conversely, identity by state (IBS) calculated between PfNF54 and each isolate shows IBS to be lower between the vaccine strain and Pf7G8 than it is between PfNF54 and any of the African strains analyzed (Supplemental information). Critically, even though the genome-wide difference between Pf7G8 and PfNF54 exceeded those between African Pf stains and PfNF54 by only about 20%, the genotypes of South America strains were fundamentally distinct from those of Pf strains in Africa (Fig. 2).

Proteome distances are defined here as the proportion of all variable non-synonymous sites that differ between each isolate and PfNF54. To investigate distance to PfNF54 based only on genomic sites with potential impact on parasite protein function, we limited the analyses to the 54,853 variable sites in which the alternative allele leads to an amino acid change. We observed the same general divergence patterns seen above for all nucleotides (Fig. 3a). Again, the multimodal distributions showed increasing proteome distances to NF54 from West to East Africa (including Madagascar). The Pf7G8 strain was the only sample that differed from PfNF54 in more than 5% of all variable amino acid-changing sites.

We do not know the peptide targets of protective immune responses after immunization with PfSPZ Vaccine, since more than 1000 Pf proteins are expressed in the sporozoite and early liver stage of the parasite[36-38]. However, based on the genomic and proteome distances of Pf7G8 from PfNF54, the number of

potential antigenic targets of protective immunity that are shared with the immunogen, PfNF54, were fewer in Pf7G8 than in all other isolates analyzed from Africa (Figs. 1 and 3).

Finally, we investigated CD8 immunome distances. Data from studies in mice and non-human primates indicate that the protective immunity induced by immunizing with radiation attenuated sporozoites is dependent on CD8+ T cells[17,39,40]. CD8+ T cells recognize ~9 amino acid peptides bound to class 1 HLA molecules[41]. To determine if the relatively large distance of Pf7G8 from PfNF54 extended to predicted CD8+ T cell epitopes, we predicted epitopes in the Pf proteome for 26 HLA alleles representative of the main allelic variants worldwide[42,43]. As with

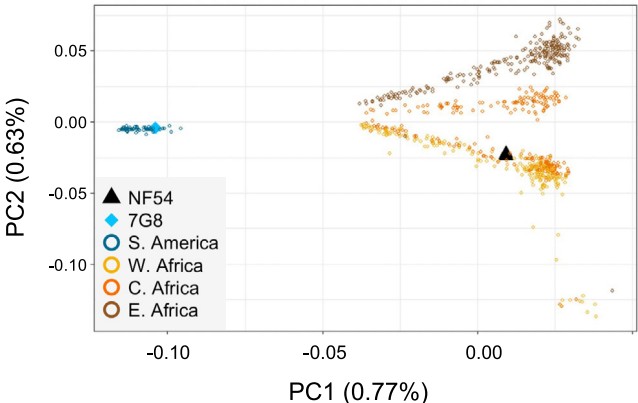

**Fig. 2 Principal component analysis (PCA) of African and South American clinical malaria samples, as well as PfNF54 and Pf7G8.** The analysis includes samples from West Africa (Mali, Guinea, and Burkina Faso; n = 234), Central Africa (Cameroon and the Democratic Republic of the Congo; n = 229), and East Africa (Kenya, Tanzania, Malawi, and Madagascar; n = 241), as well as from two South American countries (Brazil, n = 23; French Guiana, n-34) near the collection site of Pf7G8, in the Brazilian Amazon. The first PC (PC1) differentiates S. American (blue) from African strains. PC2 separates samples from the three regions in the African continent, where there is a clear differentiation between E. African (brown), Central African (orange), and W Africa (yellow). Some samples collected in Central African countries present the genomic signature of West Africa, as shown by the clustering of some orange samples with yellow samples, suggesting parasite migration through the vector or the human host. PfNF54 (black triangle) is clearly nested among W African samples, while Pf7G8 (blue diamond) clusters with those from S. America.

the whole genome/proteome data (Figs. 1 and 3a), Pf7G8 was more distant from 3D7 for predicted CD8+ T cell epitopes than were any of the isolates from Africa (Fig. 3b). It is for this reason that we think that CHMI with Pf7G8 has been as or more rigorous than field exposure; Pf7G8 is more distant from PfNF54 than are all African field isolates analyzed across the proteome and across CD8+ T cell epitopes.

Furthermore, for 42 selected pre-erythrocytic Pf antigens and vaccine candidates, non-synonymous SNPs and insertions and deletion (indels) in Pf7G8 relative to PfNF54 were common in nearly all genes, including eight in the PfCSP gene (Table 1) and added up to several hundred amino acid differences across all 42 antigens, and a different set of predicted CD8+ T cell epitopes, indicating antigenic differentiation between the two strains.

## Discussion

One of our objectives is to use PfSPZ Vaccine to protect adults with little or no previous or recent exposure to malaria, who visit areas of Africa where malaria is transmitted. These may be local travelers, such as residents of malaria-free cities, or international travelers from non-endemic countries. We cannot determine the VE of PfSPZ Vaccine in malaria naïve subjects against field-transmitted Pf malaria in Africa in a double-blind, placebo-controlled trial, because of the requirement for using chemo-prophylaxis, which is standard of care for travelers. Although non-compliance with recommended chemoprophylaxis regimens is common in travelers, under the carefully controlled conditions of a clinical trial compliance would likely be very high, thereby ensuring that the incidence of Pf infections in controls taking chemoprophylaxis during travel would be too low to achieve statistical power.

Therefore, we turned to measure VE against naturally trans-mitted malaria in the field in healthy residents of malaria endemic areas, for whom chemoprophylaxis is not the norm, and to CHMI in the malaria-naive population, where under the carefully con-trolled conditions of this clinical procedure, treatment does not need to be administered until parasitemia is detected. Thus, with both field studies in endemic populations and CHMI, endpoint determination can proceed without drug interference[14–19,22,25–28]. Using these two approaches and recognizing the limitations of small sample sizes, we have now demonstrated in several trials that the VE of PfSPZ Vaccine in semi-immune subjects against intense field transmission of heterogeneous Pf parasites over 24 weeks in the field in Africa is as good, if not better, than the VE

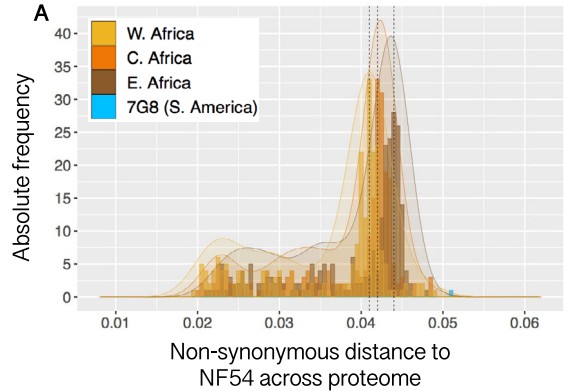

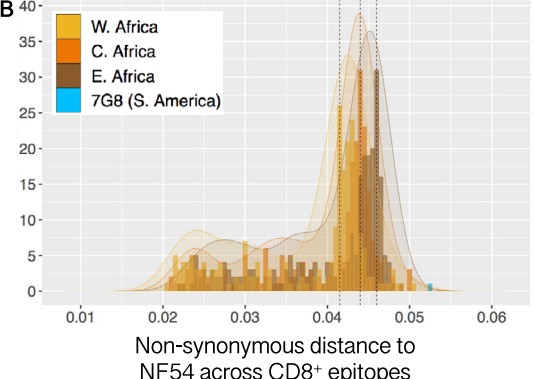

**Fig. 3 Pairwise distance in non-synonymous sites of 704 Pf African isolates and of Pf7G8 to PfNF54. a** Distance estimated across 54,853 variable, non-synonymous sites across the Pf proteome. **b** Distance estimated for 43,607 variable, non-synonymous sites that fall on predicted CD8+ T cell epitopes predicted to bind to one or more of the 26 HLA alleles analyzed. Legend as in Fig. 1. Again, the pairwise proteome difference between NF54 and the Brazilian clone Pf7G8 is larger than the distance of any African isolate to PfNF54, both across all variable non-synonymous sites (5.11%) and only in those that fall on predicted epitopes (5.27%).

**Table 1 Variants identified in Pf7G8 relative to PfNF54, in key pre-erythrocytic loci.**

| Gene ID | Product description | NF54 Protein length (AA) | Pf 7G8 Protein length (AA) | Antigenic evidence[a] | Pf 7G8 vs. NF54 AA diff.[b] | Indels[c] | Diff. in NF54 epitope Seqs[d] |
|---|---|---|---|---|---|---|---|
| PF3D7_0206900.2 | Merozoite surface protein 5 (MSP5) | 261 | 261 | A, B | 1 | 0 | 0 |
| PF3D7_0207000 | Merozoite surface protein 4 (MSP4) | 272 | 272 | A, C | 3 | 0 | 0 |
| PF3D7_0220000 | Liver stage antigen 3 (LSA3) | 1558 | 1554 | A | 48 | 6 | 48 |
| PF3D7_0304600 | Circumsporozoite protein (CSP) | 397 | 396 | A, B, C | 8 | 9 | 7 |
| PF3D7_0312400 | pfGSK3 | 440 | 440 | B | 0 | 0 | 0 |
| PF3D7_0405300 | Liver specific protein 2 (LISP2) | 1964 | 1981 | A, C | 34 | 29 | 30 |
| PF3D7_0408600 | Sporozoite invasion-associated protein 1 (SIAP1) | 984 | 984 | A | 4 | 0 | 4 |
| PF3D7_0408700 | Perforin-like protein 1 (PLP1/SPECT2) | 842 | 845 | A | 1 | 3 | 0 |
| PF3D7_0420000 | Zinc finger protein, putative | 3368 | 3312 | C | 24 | 60 | 40 |
| PF3D7_0522400 | Conserved *Plasmodium* protein, unknown function | 9307 | 9303 | C | 5 | 12 | 5 |
| PF3D7_0725100 | Conserved *Plasmodium* membrane protein, unknown function | 1576 | 1546 | C | 5 | 30 | 2 |
| PF3D7_0726400 | Conserved *Plasmodium* membrane protein, unknown function | 4944 | 4954 | C | 10 | 16 | 6 |
| PF3D7_0728600 | Zinc finger, C3HC4 type, putative | 2162 | 2186 | B | 3 | 28 | 2 |
| PF3D7_0812300 | Sporozoite surface protein 3 (SSP3) | 461 | 461 | A | 0 | 0 | 0 |
| PF3D7_0815500 | Conserved *Plasmodium* protein, unknown function | 491 | 496 | C | 0 | 5 | 0 |
| PF3D7_0826100 | E3 ubiquitin-protein ligase, putative | 8589 | 8601 | C | 8 | 42 | 7 |
| PF3D7_0828100 | Conserved *Plasmodium* protein, unknown function | 1033 | 1033 | B | 2 | 0 | 2 |
| PF3D7_0830300 | Sporozoite invasion-associated protein-2 (SIAP-2) | 388 | 388 | A | 4 | 0 | 4 |
| PF3D7_0906700 | Leucine-rich repeat protein (PfLR9) | 488 | 488 | B | 0 | 0 | 0 |
| PF3D7_1021700 | Conserved *Plasmodium* membrane protein, unknown function | 6934 | 5902 | C | 17 | 84 | 7 |
| PF3D7_1030200 | Claudin-like apicomplexan microneme protein, putative | 450 | 454 | B | 1 | 12 | 4 |
| PF3D7_1035300 | Glutamate-rich protein (GLURP) | 1233 | 1194 | C | 19 | 39 | 36 |
| PF3D7_1036400 | Liver stage antigen 1 (LSA1) | 1801 | 1630 | A, C | 147 | 171 | 86 |
| PF3D7_1121600 | Exported protein 1 (EXP1) | 162 | 162 | A | 1 | 0 | 1 |
| PF3D7_1133400 | Apical membrane antigen 1 (AMA1) | 622 | 622 | A | 28 | 0 | 24 |

**Table 1 (continued)**

| Gene ID | Product description | NF54 Protein length (AA) | Pf 7G8 Protein length (AA) | Antigenic evidence[a] | Pf 7G8 vs. NF54 | | Diff. in NF54 epitope Seqs[d] |
|---|---|---|---|---|---|---|---|
| | | | | | AA diff.[b] | Indels[c] | |
| PF3D7_1138400 | Guanylyl cyclase (GCalpha) | 4225 | 4222 | C | 2 | 7 | 2 |
| PF3D7_1147000 | Sporozoite asparagine-rich protein (SLARP) | 2940 | 2933 | A | 3 | 21 | 5 |
| PF3D7_1216600 | Cell traversal protein for ookinetes and sporozoites (CelTOS) | 182 | 182 | A | 10 | 0 | 8 |
| PF3D7_1229100 | Multidrug resistance-associated protein 2 (MRP2) | 2108 | 2073 | C | 6 | 35 | 4 |
| PF3D7_1243900 | Double c2-like domain-containing protein (PfDOC2) | 1846 | 1858 | B | 8 | 12 | 7 |
| PF3D7_1318300 | Conserved Plasmodium protein, unknown function | 1797 | 1812 | C | 3 | 29 | 2 |
| PF3D7_1325900 | Conserved Plasmodium protein, unknown function | 2746 | 2727 | C | 10 | 49 | 7 |
| PF3D7_1335900 | Thrombospondin-related anonymous protein (TRAP)/Sporozoite surface protein 2 (SSP2) | 574 | 562 | A | 19 | 12 | 15 |
| PF3D7_1342500 | Sporozoite protein essential for cell traversal (SPECT1) | 245 | 245 | A | 2 | 0 | 2 |
| PF3D7_1349300 | Tyrosine kinase-like protein (TKL3) | 1807 | 1796 | C | 2 | 35 | 11 |
| PF3D7_1365300 | Conserved Plasmodium protein, unknown function | 740 | 740 | C | 0 | 0 | 0 |
| PF3D7_1405400 | DNA mismatch repair protein, putative | 1515 | 1515 | C | 0 | 0 | 0 |
| PF3D7_1408700 | Conserved Plasmodium protein, unknown function | 7182 | 7172 | C | 7 | 24 | 18 |
| PF3D7_1438800 | Conserved Plasmodium protein, unknown function | 699 | 699 | B | 0 | 0 | 0 |
| PF3D7_1465800 | Dynein beta chain, putative | 6485 | 6494 | C | 4 | 9 | 6 |
| PF3D7_1468100 | Conservd Plasmodium protein, unknown function | 2558 | 2545 | B | 2 | 13 | 2 |
| PF3D7_1469600 | Biotin carboxylase subunit of acetyl CoA carboxylase, putative (ACC) | 3367 | 3350 | C | 8 | 17 | 19 |

[a]Three sets of genes were selected for detailed analysis coding differences: A, a list of 16 genes identified in the literature as potential pre-erythrocytic antigens; B, genes encoding proteins identified by sera from PfSPZ Vaccine vaccinees; C, genes encoding proteins identified by sera from PfSPZ-CVac vaccinees[31].
[b]Number of amino acid residue differences between the NF54 protein and its 7G8 ortholog, including insertions and deletions (indels).
[c]Cumulative length of insertions and deletions (indels), measured in amino acid residues, in 7G8 relative to NF54.
[d]Number of NF54 amino acid residues in predicted CD8+ T cell epitopes that differ in the 7G8 ortholog (amino acid residue differences or deletions).

in malaria-naïve subjects immunized in the US with the identical immunization regimen of PfSPZ Vaccine against heterologous CHMI with Pf7G8 at 24 weeks after the last dose of vaccine[17,28,34,35].

The antibody responses to PfSPZ Vaccine in semi-immune subjects in Africa were 9–28 times lower than the antibody responses in previously malaria naïve subjects in the US. Although anti-sporozoite antibody responses may not be causally related to VE, the lower response may indicate generally lower immunogenicity of the vaccine in previously malaria-exposed subjects. We have reported that adults in Tanzania make T cell responses to the same immunization regimen of PfSPZ Vaccine more than five-fold lower than do adults in the US[21]. Due to better vaccine take, malaria-naïve travelers immunized with PfSPZ Vaccine may therefore exhibit better VE against circulating Pf strains in the field than malaria-exposed residents. For this reason, VE against heterologous CHMI with Pf7G8 should provide a conservative prediction of VE in malaria-naïve travelers, whose immune reponses are not disadvantaged by prior malaria exposure.

Data from genomic sequencing indicate that Pf7G8 is more distinct from the vaccine strain (PfNF54) than are all 704 Pf isolates from East, Central, and West Africa that we analyzed (Figs. 1 and 2). This pattern holds true when comparisons are restricted to sites encoding predicted CD8+ T cell epitopes (Fig. 3). Therefore, due to genetic divergence, Pf7G8 shares fewer identical epitopes with the vaccine strain than do Pf isolates from Africa. We hypothesize that this difference explains why it is harder to protect against Pf7G8 than against Pf isolates from Africa, and why CHMI with Pf7G8 provides a stringent test of PfSPZ Vaccine efficacy.

Efforts to bridge from VE measured against CHMI to field protection must account for the heterogeneity of field isolates and, until our data, the characteristics of the CHMI strains needed to evaluate VE for global travelers and the basis for their selection have been unclear. This knowledge gap was pointed out in a 2017 paper from the US Food and Drug Administration[44], which cautioned that protective immune responses may be strain or variant specific, limiting the predictive value of CHMI, especially when conducted with homologous parasites. The authors hypothesized that "the high degree of genetic and antigenic diversity among *P. falciparum* isolates across and within geographical regions, evidence of protection in CHMI for multiple strains will likely be needed to support the effectiveness of a malaria vaccine against diverse strains."

The data reported herein address this knowledge gap, providing a bridge from CHMI to field protection now supported by two independent clinical data sets and a plausible underlying mechanism. In the absence of field data from malaria-naives, which we intend to obtain during Phase 4 (post marketing) surveillance, heterologous CHMI using a strain that is more divergent from the vaccine strain than any Africa strain should provide appropriate data to support vaccine licensure for travelers to Africa. In addition, the VE data will provide healthcare professionals in countries and areas of countries where malaria is not transmitted with sound information upon which to make recommendations for travelers to endemic areas. As more areas within malaria endemic countries are now malaria free (such as capital cities), individuals growing up in these areas have no more naturally acquired immunity to malaria than do non-immunes growing up in the US or EU, extending the applicability of findings from heterologous CHMI to this population when undertaking within-country travel. Ideally, a malaria vaccine would protect against all Pf isolates in Africa. However, since there are so many strains of Pf, this is impossible to assess in clinical trials. Since Pf7G8 is more genetically distant from the

vaccine strain of Pf than virtually all Pf strains in Africa, we believe that heterologous CHMI with Pf7G8 can act as a surrogate for field trials in all regions of Africa, and will provide a rigorous estimate of VE against naturally transmitted, heterogeneous Pf in the field. If this can be further substantiated, it should allow a reduction in the number of field studies in endemic populations needed to support the clinical development of new malaria vaccine candidates.

## Methods

**Vaccine trials of PfSPZ Vaccine in the US and Mali.** We assessed the same immunization regimen of PfSPZ Vaccine in the US[17,28] and Mali[34,35]. The trials were registered on Clinical Trials.gov (https://clinicaltrials.gov) (Trial 1 in US: NCT02215707; Trial 2 in US: NCT02601716; Trial 3 in Mali: NCT01988636; Trial 4 in Mali: NCT02627456). The trials were conducted according to Good Clinical Practices and the International Conference on Harmonization (ICH) and institutional procedures and guidelines and approved by the appropriate institutional review boards (IRBs) in the United States (US) and Mali (Trial 1 in the US[17] – IRB of Walter Reed Army Institute of Research; Trial 2 in the US[28] – IRBs of University of Maryland Baltimore and Naval Medical Research Center; Trials 3[34] and 4[35] in Mali – ethics review board of Faculté de Médecine de Pharmacie et d'OdontoStomatologie [FMPOS], Bamako), IRB of US National Institute of Allergy and Infectious Diseases (NIAID), National Institutes of Health [NIH], and the Mali national regulatory authority). All trials were conducted under US FDA IND #14826. Written informed consent was obtained from all subjects prior to inclusion in the study. In Mali, in addition to individual consent, each participating village provided community permission.

**Vaccine.** Sanaria® PfSPZ Vaccine is composed of radiation attenuated (metabolically active, non-replicating) aseptic, purified, vialed, cryopreserved PfSPZ. The parasites are from the NF54 strain of Pf, which was isolated in the Netherlands[13] and is presumed to be of West African origin[31]. The proteome of NF54 was shown to be virtually identical to that of the NF54-derived Pf reference clone 3D7, with only 15 of all ~5000 single copy protein-coding genes with amino acid residue differences between the two[31]. The vaccine is administered as a 0.5 mL injection by direct venous inoculation (DVI) through a 25-gauge needle.

**CHMI.** CHMI in the US trials[17,28] was done by the bite of 5 *Anopheles stephensi* mosquitoes infected with Pf7G8, which is a clone of a Pf strain (IMTM22) isolated in Brazil[29,30,45]. Research subjects were followed for 28 days and were considered protected if no parasitemia developed. VE following CHMI was assessed by proportional analysis (1-risk ratio, where risk ratio was the proportion of vaccinees infected divided by the proportion of controls infected).

**Field Trials.** In field trials, subjects received a curative dose of an antimalarial approximately 2 weeks before the first and last dose of PfSPZ Vaccine. They were then followed actively every two weeks for 26 weeks by assessment of thick blood smears (TBS) for the presence of Pf parasites. If at any time during follow-up the subjects had symptoms that could be caused by malaria (e.g., fever, headache), they were instructed to come to the study clinic for TBS assessment. VE was assessed by time to event analysis (1-hazard ratio) and by proportional analysis (1-risk ratio)[34,35].

**Genomic data and genotyping approach.** Whole genome shotgun sequence (WGS) data for PfNF54, the strain in PfSPZ Vaccine, with origin in West Africa, and for Pf7G8, the strain for heterologous CHMI, originating from Brazil, were generated previously[31]. WGS data, obtained with or without selective whole genome amplification (sWGA), for 704 Pf clinical isolates from Africa, including West Africa (Guinea, $n = 124$; Mali, $n = 54$ and Burkina Faso, $n = 56$), central Africa (Cameroon, $n = 122$; Democratic Republic of Congo, $n = 107$) and East Africa (Kenya, $n = 52$; Tanzania, $n = 68$; Malawi, $n = 103$; Madagascar, $n = 18$) were used in the analyses (Supplementary Data 1). Finally, Pf clinical isolates from South America ($n = 85$, including 22 from Brazil) and Southeast Asia ($n = 278$) were included only in the joint sequence variant identification step, including single nucleotide polymorphism (SNP) calls (see below), to ensure comprehensive and accurate identification of variable genomic sites, particularly in Pf7G8. WGS data for isolates from Brazil, Mali and Malawi were generated in house, at the Institute for Genome Sciences[31], and those from Peru, Colombia, French Guiana, Burkina Faso, Cameroon, Democratic Republic of Congo, Guinea, Kenya, Tanzania, Madagascar, and some from Cambodia were downloaded from NCBI's Short Read Archive (Supplementary Data 1). Data processing and sequence variant calling were conducted as previously described[31]. Briefly, reads were aligned to the Pf 3D7 reference genome (PlasmoDBv24) using bowtie2 (v2.2.4)[46], and resulting bam files processed according to GATK's Best Practices documentation[47]. Joint SNP calling was done using Haplotype Caller (v4.0), and bi-allelic sites were selected. To capture the predominant clone in polyclonal infections, the major allele was called if supported by >70% of reads, and otherwise coded as missing. The following stringent

hard filter was also applied: DP < 12 || QUAL < 50 || FS > 14.5 || MQ < 20. Finally, variant sites were filtered out if the minor allele was present in fewer than three samples or when missing genotype values were present in 10% or more of all samples. SNPs were classified as synonymous (silent) and non-synonymous (amino acid-altering) using snpEFF (v4.3t)[48].

**CD8 T cell epitope prediction**. MHC class I epitopes eight to eleven amino acids in length were predicted in the proteome of Pf3D7 for 26 HLA alleles representative of the main allelic variants worldwide[42], using NetMHCpan (v4.0a)[49], and those predicted to be strong binders were kept. The reunion of all epitope for each of the Pf 5546 protein-coding genes was mapped back to the Pf3D7 reference genome assembly to determine their coordinates, and the intersection of these genomic coordinates with the set of non-synonymous sites that differ between 3D7 and each sample was identified.

**Genetic, proteome, and epitome distances**. SNPs were called for all samples and strains against the reference 3D7 genome as described above, and genomic sites variable across the dataset of all samples were identified. Variable sites not called for Pf7G8 (<0.2% of all variable sites) were eliminated. For each sample, genetic distance to NF54 was calculated as the proportion, $p$, of variable sites in the data partition that differed between it and NF54. Three data partitions were considered: (i) all variable positions, (ii) non-synonymous (NSYN) sites (those where change between each isolate and Pf3D7 led to a difference in amino acid), and (iii) NSYN sites in predicted Pf3D7 epitopes. For each data partition, the distribution of distances to NF54 were presented as a histogram, with approximation to a continuous distribution generated by Gaussian kernel density estimate using the ggplot2 package in R[50], and kernel density estimate calculated as 0.0015× number of samples.

In addition, IBS, a metric of genetic similarity that is essentially the complement of $p$ distance, was estimated using all sites genotyped in both PfNF54 and the sample in each pairwise comparison (instead of limiting the comparisons to sites genotyped in Pf7G8), using PLINK v1.90[51]. IBS values were calculated based on variable sites across all samples genotyped.

**Principal components analysis (PCA)**. SNPs used to estimate genetic distances were further assessed for linkage disequilibrium (LD), to eliminate undue weight from genetically-linked sequence variants. The dataset was pruned to eliminate SNPs in LD with any other SNPs (defined by $r^2 \geq 0.3$) within 500 Kb in the same chromosome. LD pruning resulted in a dataset of 33,256 SNPs. A PCA was conducted using this trimmed SNP set as input. Analyses were conducted with SNPRelate package in R, using functions snpgdsLDpruning and snpgdsPCA[52].

**Reporting summary**. Further information on research design is available in the Nature Research Reporting Summary linked to this article.

## Data availability

The genome of the reference Pf 3D7 strain and respective structural annotation were obtained from PlasmoDB (release-24). All whole genome sequence datasets used in this study were downloaded through the NCBI's Sequence Read Archive (SRA) database. Accession IDs are available in Supplementary Data 1. For the clinical trials[17,28,34,35], the data supporting the findings of studies are available within the Articles and their Supplementary Information and from the corresponding authors upon reasonable request, and after execution of inter-institutional human data sharing agreements.

## Code availability

All computer programs used in the analysis of these data are referenced in the "Methods" section.

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

## Acknowledgements

We thank Tomás Harris for algorithm development. The work at the University of Maryland was funded in part by the National Institutes of Health (NIH) awards U19AI110820 (J.C.S.) and R01AI141900 (J.C.S.). The work at Sanaria was supported by NIAID, NIH SBIR grants 5R44AI058375-10 (S.L.H.) and 5R44AI055229-11 (S.L.H.), and by DoD Award/Contract W81XWH1420011 (S.L.H.). The work at Naval Medical Research Center was also supported/funded by work unit numbers A1217 and A1237 (J.E.E.). Authors S.A.H. and P.E.D. are supported by the Intramural Research Program of the National Institute of Allergy and Infectious Diseases, National Institutes of Health. The study protocol for JWMRP2 was approved by the Naval Medical Research Center Institutional Review Board in compliance with all applicable federal regulations governing the protection of human subjects. J.E.E. was a military service member. This work was prepared as part of her official duties. The views expressed in this article reflect the results of research conducted by the authors and do not necessarily reflect the official policy or position of the Department of the Navy, Department of Defense, nor the United States Government. Title 17 U.S.C. 105 provides that copyright protection under this title is not available for any work of the United States Government. Title 17 U.S.C. §101 defines a U.S. Government work as work prepared by a military service member or employee of the U.S. Government as part of that person's official duties.

## Author contributions

J.C.S and S.L.H. conceived and designed the experiment, and interpreted results. J.C.S., A.D., and K.A.M. defined data analysis plan, developed algorithms, gathered, and analyzed the data. J.C.S., S.L.H., and T.L.R. drafted the manuscript. M.S.S., J.E.E., S.A.H., K.E.L., B.M., P.G.K., and P.E.D. conducted the clinical trials that provided the foundation for conducting the experiments, and T.L.R. supervised these clinical trials for the sponsor. T.M. directed all interaction with regulatory authorities. B.K.L.S. supervised the production of parasites. T.M. and B.K.L.S. provided ideas for integration of laboratory and field results. All authors read, commented on, and approved the final version of the manuscript.

## Competing interests
