## [Peer Review File · Nature Communications]

Plasmodium falciparum 7G8 challenge provides conservative prediction of efficacy of PfNF54-based PfSPZ Vaccine in AfricaReviewers' Comments:

Reviewer #1:

Remarks to the Author:

Summary: The authors showed that vaccine efficacy assessment of PfSPZ in adults in the US against a Brazilian parasite-based (Pf7G8) heterologous CHMI was more stringent than assessment with homologous CHMI. They concluded that while the challenge strain (Pf7G8) was genetically divergent from PfSPZ the vaccine strain (PfNF54), it (Pf7G8-based CHMI) provided a reliable estimate of vaccine PfSPZ vaccine protection. This report is essential as it suggests that Pf7G8-based CHMI is a potential option for current largely expensive and onerous PfSPZ vaccine trials to optimize dosage regimen. The study design is quite clear and the methods are appropriate and interoperable. However, the manuscript includes some erroneous generalizations which the authors can address before publication.

Major revisions
Title: Proteomic analyses were not exactly carried out in this study. Authors only analyzed nonsynonymous mutations as proxy for estimating protein-coding genes. This is clearly not a conventional proteomics approach. One wonders why protein microarray, Edman sequencing or chromatography-based approach etc was not adopted. However, given that the study is a straightforward genetic diversity analysis, we recommend that the authors stick to the broader term "genome" in place "proteome" which was not comprehensively described in this study. Please see suggested title below. Furthermore, how generalizable is immunogenicity to this West African strain considering high level of antigenic variability across different endemic settings in Africa? The title of this manuscript seems to assume antigenic homogeneity across Africa. It may seem useful to focus the title on "West Africa or even Mali" rather than blanket "Africa".

New suggested title: "Genome, proteome, and immunome data explain why a six-6 month controlled human malaria infection with sporozoites of the Pf7G8 clone of Plasmodium falciparum is a rigorous predictor of the efficacy of the PfNF54-based PfSPZ Vaccine in West Africa"

Main: Line 83-85 - While there is clustering along continental regions, genetic heterogeneity of Pf within Africa also occurs. For instance, Amambua-Ngwa et al. (Science 365 (6455), 813-816) demonstrated population structuring in Africa with a distinctive population in Ethiopia. Therefore, it may be erroneous to assume African parasites are genetically homogenous.

Line 106 - This is a sweeping statement. Were all Pf strains in Africa investigated in this study? Definitely not. The 709 isolates analyzed do not make for all the Pf strains in Africa.

Results

Line 184 - I understand the authors did some nucleotide based analysis. However, there are more scientific ways of genetic correlations to ascertain divergence. For instance, identity-by-descent (IBD) analysis could provide more confidence in the genetic relatedness or divergence claims. Authors are encouraged to carry out IBD analysis.

Line 202 - Proteomic analyses were not exactly carried out in this study. Authors only analyzed nonsynonymous mutations as proxy for estimating protein-coding genes. This is clearly not a conventional proteomics approach. Authors may want to change the title of this section to "Genetic divergence across protein-coding regions."

Minor revisions

Main: Lines 74 and 77 - Add references. Line 94 - Replace "South American" with "Brazilian".

Line 100 - Break the sentence "...divergence, and found..." to "...divergence. We found...".

Line 107 - Change "By extending our argument, a general approach..." to "By extending our argument, a similar approach..."

Results: Line 211 - "However, based on the genomic/proteomic distance..." can be replaced by "However, based on the genomic distance..."

Discussion: Line 242: Please provide a reference. Lines 270, 271 - No need to refer to figures in Discussion. This was already highlighted in Results. Line 272 - "hypothesize" to "hypothesized". Line 277 - "has" to "had". Line 299 - Please provide reference after "...clinical trials".

Reviewer #2:

Remarks to the Author:

Establishing the efficacy of a malaria vaccine in travelers is a challenge because of the relative infrequency of malaria in travelers, except in special circumstances, and hence the need for a very large and expensive trial. Thus, the hypothesis set forward in this paper that postulates that if a vaccine based on an African strain has have a high level of protection in a challenge study in potential travelers with a parasite strain that is widely divergent genetically and structurally from strains prevalent across sub-Saharan Africa, there is a good chance that it would protect travelers to these areas is an important one.

The paper presents strong evidence that the Brazilian strain Pf7G8 is genetically widely separated from nearly all African strains and is thus suitable for challenge studies needed to validate this hypothesis. The paper is well written and easy to follow and I have only a few comments and suggestions.

Main comment

My main comment is that the study described provides strong evidence that the Pf7G8 strain from Brazil is genetically very different from a large panel of isolates obtained from across sub-Saharan Africa and has important structural differences in key antigens likely to play an important role in protective immunity, an essential finding for the hypothesis. In contrast, the clinical support for the hypothesis, better protection in Malian subjects exposed to African isolates than the protection seen in challenges studies with a heterologous strain in the US is on less solid grounds because of small numbers in the challenge studies. Confidence levels on efficacy values are not given for the efficacy studies. The small number of subjects in the in the challenge and hence the need for caution in comparing results between studies this needs to be noted in the discussion.

Minor comments.

p.3, l.45. Burden of malaria. There is a new WHO World Malaria Report for 2020 data so these figures, which are worse than the ones presented, could be updated.

p.3, l55. Malaria in travelers. Like wise there are updated statistics on malaria in the UK – there were 1719 cases and 15 deaths. Because the incidence of malaria in travelers is very low, making phase 3 efficacy trial very difficult as the authors point out, cost effectiveness will become an important issue in relation to vaccines for travelers but this is not an area for consideration in this paper.

p.6.l. 121 The calculation leading to an 8% VE shown in brackets, is not easy to understand – where does the multiplication come from?

p.11, l.230. Trials in special groups. I agree that evaluating efficacy in routine travelers would be very challenging but there may be special circumstances in which this might be possible, for example in troops exposed to a high level of infection in whom not taking prophylaxis might be possible if there was strong background information on the efficacy of the vaccine. chemoprophylaxis is not given given routinely to the 20 million workers going to work in sub-Sharan Africa each year.

Point-by-point response to the reviewers' comments

REVIEWER COMMENTS

Reviewer #1 (Remarks to the Author):

Summary: The authors showed that vaccine efficacy assessment of PfSPZ in adults in the US against a Brazilian parasite-based (Pf7G8) heterologous CHMI was more stringent than assessment with homologous CHMI. They concluded that while the challenge strain (Pf7G8) was genetically divergent from PfSPZ the vaccine strain (PfNF54), it (Pf7G8-based CHMI) provided a reliable estimate of vaccine PfSPZ vaccine protection. This report is essential as it suggests that Pf7G8-based CHMI is a potential option for current largely expensive and onerous PfSPZ vaccine trials to optimize dosage regimen. The study design is quite clear and the methods are appropriate and interoperable. However, the manuscript includes some erroneous generalizations which the authors can address before publication.

Major revisions

Title: Proteomic analyses were not exactly carried out in this study. Authors only analyzed nonsynonymous mutations as proxy for estimating protein-coding genes. This is clearly not a conventional proteomics approach. One wonders why protein microarray, Edman sequencing or chromatography-based approach etc was not adopted. However, given that the study is a straightforward genetic diversity analysis, we recommend that the authors stick to the broader term “genome” in place “proteome” which was not comprehensively described in this study.

Response: We understand the reviewer's comment, and we respectfully disagree with its principle. The reviewer is correct in that our proteomics (or “immunomics”, for that matter) analyses did not rely on traditional experimental protein or epitope data – which was not doable since only whole genome sequencing data is available for these hundreds of Pf isolates. Accordingly, we changed “data” for “analyses” in the title. However, our analyses did focus on three separate data partitions, namely all variable sites in the genome, only variable non-synonymous sites in protein-coding sequences and only those in potential T cell epitopes. We consider it important that the title reflects that fact. Finally, PfNF54 and Pf7G8 were compared with hundreds of samples in West, Central and East Africa, and so the results of the analyses are in fact applicable to those three regions.

Please see suggested title below. Furthermore, how generalizable is immunogenicity to this West African strain considering high level of antigenic variability across different endemic settings in Africa?

The title of this manuscript seems to assume antigenic homogeneity across Africa. It may seem useful to focus the title on "West Africa or even Mali" rather than blanket "Africa".

New suggested title: “Genome, proteome, and immunome data explain why a six-6 month controlled human malaria infection with sporozoites of the Pf7G8 clone of Plasmodium falciparum is a rigorous predictor of the efficacy of the PfNF54-based PfSPZ Vaccine in West Africa”

Response: We appreciate the Reviewer's comment but point out that 7G8 is highly distant from parasites all over Africa. We prefer to leave the title as is, but if the editors believe it appropriate, we can change from “Africa” to “West Africa.”

Main: Line 83-85 - While there is clustering along continental regions, genetic heterogeneity of Pf within Africa also occurs. For instance, Amambua-Ngwa et al. (Science 365 (6455), 813-816) demonstrated

population structuring in Africa with a distinctive population in Ethiopia. Therefore, it may be erroneous to assume African parasites are genetically homogenous.

Response: We agree with the reviewer and have altered our explanation. For example, in this paper and in our published paper (ref 29, Moser et al.), we report differences between East, West, and Central Africa. We are well aware of the Pf genetic variation within the African continent and have edited accordingly.

Line 106 - This is a sweeping statement. Were all Pf strains in Africa investigated in this study? Definitely not. The 709 isolates analyzed do not make for all the Pf strains in Africa.

Response: The response has been edited to refer only to the 704 isolates we assessed, and we have edited accordingly throughout the manuscript.

Results

Line 184 - I understand the authors did some nucleotide based analysis. However, there are more scientific ways of genetic correlations to ascertain divergence. For instance, identity-by-descent (IBD) analysis could provide more confidence in the genetic relatedness or divergence claims. Authors are encouraged to carry out IBD analysis.

Response: We respectfully disagree that IBD is the desired distance metric. In fact, what is critical here is sequence similarity *_per se_* between strains in endemic areas and PfNF54, the vaccine strain, and not whether that sequence similarity was arrived at by descent (strains recently acquired the same sequence variant from a common ancestor with PfNF54, IBD) or whether they just share variants by state (genomic variants commonly present in Pf natural populations, IBS). We very much doubt that there are any strains from natural populations with a significant proportion of the genome IBD with PfNF54, which was isolated 40 years ago and has been in the lab ever since. Nonetheless, we calculated IBS (the complement of the p distances originally estimated) and present the results in supplemental figure S4.

Line 202 – Proteomic analyses were not exactly carried out in this study. Authors only analyzed nonsynonymous mutations as proxy for estimating protein-coding genes. This is clearly not a conventional proteomics approach. Authors may want to change the title of this section to “Genetic divergence across protein-coding regions.”

Response: We respectfully disagree. Given that the proteome of an organism is the set of all proteins encoded in its genome, proteome distances are the amino acid residue differences accumulated across those proteins. We altered the text to make this clear.

Minor revisions

Main: Lines 74 and 77 – Add references. Line 94 – Replace “South American” with “Brazilian”.

Response: Done.

Line 100 – Break the sentence “...divergence, and found...” to “...divergence. We found...”.

Response: Done.

Line 107 – Change “By extending our argument, a general approach...” to “By extending our argument, a similar approach...”

Response: *Done. We also changed approach to rationale.*

Results: Line 211 – “However, based on the genomic/proteomic distance...” can be replaced by “However, based on the genomic distance...”

Response: *We replaced “proteomic distances” with “proteome differences”, and the latter are now clearly defined.*

Discussion: Line 242: Please provide a reference.

Response: *Done.*

Lines 270, 271 - No need to refer to figures in Discussion. This was already highlighted in Results.

Response: *We think it helps the reader to direct them to the figures. If the editors prefer us to take these references out, we will.*

Line 272 – “hypothesize” to “hypothesized”.

Response: *We leave this grammatical point to the editors. We have reported the results in the past tense, but here, based on the results, we are hypothesizing in the present tense.*

Line 277 – “has” to “had”.

Response: *We cannot find the “has” being referred to at line 277.*

Line 299 – Please provide reference after “...clinical trials”.

Response: *We are not quite sure what the reviewer is referring to. We have now put in the actual link for ClinicalTrials.gov. Is that what the reviewer was referring to?*

Reviewer #2 (Remarks to the Author):

Establishing the efficacy of a malaria vaccine in travelers is a challenge because of the relative infrequency of malaria in travelers, except in special circumstances, and hence the need for a very large and expensive trial. Thus, the hypothesis set forward in this paper that postulates that if a vaccine based on an African strain has have a high level of protection in a challenge study in potential travelers with a parasite strain that is widely divergent genetically and structurally from strains prevalent across sub-Saharan Africa, there is a good chance that it would protect travelers to these areas is an important one.

The paper presents strong evidence that the Brazilian strain Pf7G8 is genetically widely separated from nearly all African strains and is thus suitable for challenge studies needed to validate this hypothesis. The paper is well written and easy to follow and I have only a few comments and suggestions.

Main comment

My main comment is that the study described provides strong evidence that the Pf7G8 strain from Brazil

is genetically very different from a large panel of isolates obtained from across sub-Saharan Africa and has important structural differences in key antigens likely to play an important role in protective immunity, an essential finding for the hypothesis. In contrast, the clinical support for the hypothesis, better protection in Malian subjects exposed to African isolates than the protection seen in challenge studies with a heterologous strain in the US is on less solid grounds because of small numbers in the challenge studies. Confidence levels on efficacy values are not given for the efficacy studies.

Response: *The confidence levels are included in 3 of the 4 publications (not in Lyke 2021).*

For immunization regimen #1 which had an apparent difference in VE by risk ratio analysis (10% vs 29%), we have included the following statement on line 151.

“Although the VE by I-risk ratio analysis in Mali (29%) was higher than in the US (10%), the 95% confidence intervals overlapped and thus, the differences were not statistically significant (18, 33). This was in part due to the small sample size in the CHMI study.”

We have not included the actual confidence intervals which were in the publications, Epstein (2017, US): 10% (-35.8 to 45.6) vs Sissoko (2017, Mali): 28.8% (8.2 to 47.2) We could include these if the editors think it is important.

For immunization regimen #2, since VE by I-risk ratio analysis was 23% in the US and 24% in Mali, we didn't think it necessary to include the statement for this set of data, as it is obvious that there is no difference. Let us know if you would like it included. Also, the 95% confidence interval, which was not included in the manuscript, was (-13 to 50).

Furthermore, in both Mali trials the VE by I-risk ratio and by I-hazard ratio was significant, but in both US trials VE was not significant.

The small number of subjects in the in the challenge and hence the need for caution in comparing results between studies this needs to be noted in the discussion.

Response: *A qualifying clause has been added to the Discussion (lines 292-293).*

Minor comments.

p.3, 1.45. Burden of malaria. There is a new WHO World Malaria Report for 2020 data so these figures, which are worse than the ones presented, could be updated.

Response: *This has now been updated.*

p.3, 1.55. Malaria in travelers. Like wise there are updated statistics on malaria in the UK – there were 1719 cases and 15 deaths. Because the incidence of malaria in travelers is very low, making phase 3 efficacy trial very difficult as the authors point out, cost effectiveness will become an important issue in relation to vaccines for travelers but this is not an area for consideration in this paper.

Response: *Numbers and reference updated.*

p.6.1. 121 The calculation leading to an 8% VE shown in brackets, is not easy to understand – where does the multiplication come from?

Response: *We have changed the section to state 10%; 1/10 protected at 6 months. The 8% and calculations have been eliminated.*

p.11, 1.230. Trials in special groups. I agree that evaluating efficacy in routine travelers would be very challenging but there may be special circumstances in which this might be possible, for example in troops exposed to a high level of infection in whom not taking prophylaxis might be possible if there was strong background information on the efficacy of the vaccine. chemoprophylaxis is not given routinely to the 20 million workers going to work in sub-Saharan Africa each year.

Response: *We are planning to do a trial in Indonesian military beginning in March 2022. However, we have not had the opportunity to do such a trial in Africa, although we have been considering it for decades. We will investigate the workers going to Africa, but here again, we have to deal with the tremendously increased costs of doing such trials.*

Reviewers' Comments:

Reviewer #1:

None

Reviewer #2:

Remarks to the Author:

Thank you for addressing my comments and suggestions satisfactorily.